# Long-Term Outcomes of Laparoscopic Liver Resection for Centrally Located Hepatocellular Carcinoma

**DOI:** 10.3390/medicina58060737

**Published:** 2022-05-30

**Authors:** Hyo Jun Kim, Jai Young Cho, Ho-Seong Han, Yoo-Seok Yoon, Hae Won Lee, Jun Suh Lee, Boram Lee, Yeongsoo Jo, Meeyouong Kang, Yeshong Park, Eunhye Lee

**Affiliations:** Department of Surgery, Seoul National University Bundang Hospital, Seoul National University College of Medicine, Seoul 13620, Korea; hyojun.kim91@gmail.com (H.J.K.); hanhs@snubh.org (H.-S.H.); yoonys@snubh.org (Y.-S.Y.); lansh@hanmail.net (H.W.L.); rudestock@gmail.com (J.S.L.); boramsnubhgs@gmail.com (B.L.); 82588@snubh.org (Y.J.); rime0317@gmail.com (M.K.); yeshong.park@gmail.com (Y.P.); eunhye.lee531@gmail.com (E.L.)

**Keywords:** laparoscopy, laparoscopic liver resection, centrally located HCC, indication

## Abstract

*Background and Objectives:* The feasibility of laparoscopic liver resection (LLR) for centrally located hepatocellular carcinoma (cHCC 1 cm of the hilum, major hepatic veins, and inferior vena cava) is still controversial. This study aims to evaluate the feasibility and safety of LLR for cHCC and compare the perioperative outcomes with those of open liver resection (OLR). *Materials and Methods:* This retrospective study included 110 patients who underwent LLR (n = 59) or open liver resection (OLR) (n = 51) for cHCC between January 2004 and September 2018. LLR group was divided into the following two subgroups according to the date of operation: Group 1 (n = 19) and Group 2 (n = 40), to account for the advancement in the laparoscopic techniques. *Results:* No mortality within 3 months was observed. There were no significant differences in operation time (285 vs. 280 min; *p* = 0.938) and postoperative complication rate (22.0% vs. 27.5%; *p* = 0.510) between both groups. However, intraoperative blood loss (500 vs. 700 mL; *p* < 0.001), transfusion rate (10.2% vs. 31.4%; *p* = 0.006), and hospital stay (6 vs. 10 days; *p* < 0.001) were significantly lower in the LLR group than in the OLR group. In the LLR group, Group 2, showed a shorter hospital stay than Group 1 (6 vs. 8 days; *p* = 0.006). There were improvements in the operation time (280 vs. 360 min; *p* = 0.036) and less intraoperative blood loss (455 vs. 500 mL; *p* = 0.075) in Group 2. *Conclusions:* We demonstrated that LLR can be safely performed in highly selected patients with cHCC.

## 1. Introduction

Liver disease is highly prevalent in Asian countries including Korea and liver cancer is known to be the third leading cause of cancer-related death [1,2]. The majority of the primary liver cancer is composed of hepatocellular carcinoma (HCC). Surgical resection of the liver is considered the first-line treatment option for HCC if indicated [3,4]. Laparoscopic liver resection (LLR) is becoming widely accepted as a feasible option when performing liver resection [5,6]. Previous studies have reported less blood loss, shorter operation time, and shorter hospital stay for LLR compared with open liver resection (OLR) [7,8,9]. Laparoscopic minor resection and left lateral sectionectomy are now considered standard procedures according to the recent consensus meeting held in Morioka, Japan [10]. However, the feasibility and safety of LLR for centrally located HCC (cHCC) is still controversial since they are usually located very close to the liver hilum, major hepatic veins, or inferior vena cava (IVC) [11]. Therefore, laparoscopic resection for such tumors is often considered a contraindication due to possible injuries to major vascular structures and difficulty in controlling bleeding [12].

Accumulation of experience and development of novel instruments have enabled surgeons to perform LLR for patients with cHCC. Several previous small case series studies, including those from our center, reported that LLR can be performed safely in the selected patients with cHCC [12,13,14,15]. However, no studies have compared perioperative and long-term outcomes between LLR and OLR for HCC located at the central segments of the liver. Thus, this study aimed to evaluate the feasibility and safety of LLR for cHCC and compare the perioperative outcomes with those of OLR. The LLR group was further divided into two subgroups to compare the outcomes of the recent advancements in laparoscopic techniques and devices.

## 2. Materials and Methods

### 2.1. Patients

The medical records of 634 patients who underwent LLR for HCC at our hospital between January 2004 and September 2018 were retrospectively reviewed. Among them, 59 patients with cHCC were analyzed. The perioperative outcomes of these patients were compared with those of 51 patients who underwent OLR for tumors at a similar location during the same period. In the LLR group, there were two open conversion cases because of intraoperative bleeding and difficulty in dissecting laparoscopic lymph nodes, and they were excluded from the analysis.

The patients in the LLR group were further divided into two groups, Group 1 (n = 19) and Group 2 (n = 40), who underwent LLR before and after 2015, respectively, to evaluate the outcomes in terms of the advancement of the laparoscopic techniques and devices [16,17].

### 2.2. Definitions

cHCC is defined as HCC within 1 cm or less of major vascular structures, including the hilum, major hepatic veins, and inferior vena cava (IVC). The locations of the tumors and the distances between the tumors and the major vascular structures were measured on the basis of preoperative computed tomography scans [12].

All of the cirrhotic patients had histologically confirmed liver cirrhosis, determined by a pathologist. The severity of complications was graded using the Clavien–Dindo classification [18]. Postoperative complications were defined as those occurring up to 30 days after surgery. Postoperative mortality was defined as death within 90 days after the operation. Overall survival was calculated from the date of the operation to the date of death or the last follow-up. The Brisbane 2000 terminology was used to define liver resection [19].

### 2.3. Surgical Techniques

Although indications for LLR were similar to those of OLR, LLR was not considered for a tumor size greater than 5 cm in diameter and tumors invading or adjacent to the main portal pedicle or IVC [12].

Surgical procedures for LLR at our center have been described in several other studies [16,20,21]. Details of each resection to be performed were decided based on the location of the tumors. Before the operation, the location of the tumors and their anatomical relationship with major vascular structures were analyzed using preoperative radiological imaging. Intraoperative ultrasonography was used for the modification of the resection during the operation. Anatomical liver resection was preferred if indicated. However, subsegementectomy was performed for the tumor located at the Spiegel lobe of the caudate lobe.

The details of the laparoscopic procedures are as follows. Under general anesthesia, the patients were positioned in a lithotomy position and tilted to a 30° reverse Trendelenburg position. The patients were placed in a French position and semi-lateral decubitus with their legs spread. We placed a 12 mm camera port at the sub-umbilical position, and pneumoperitoneum was subsequently established, maintaining the pressure below 13 mmHg. The two main working ports, either 11 or 12 mm, were inserted at the subcostal area meeting with the midclavicular line and epigastric area. An additional 5 mm port was inserted at the subcostal area meeting with the anterior axillary line. Furthermore, two more ports were meticulously placed in the 7th and 9th intercostal spaces [17]. Flexible laparoscope and intraoperative ultrasound were extensively utilized to identify the exact tumor location and check the resection margin. For anatomical resection, the Glissonean pedicle or supplying portal pedicle of the segment of section to be resected was controlled beforehand. Laparoscopic Pringle’s maneuver was performed by clamping the hepatoduodenal ligament with the ends of the umbilical tape passing through the long tube. Each clamping time was restricted to a maximum of 15 min to avoid ischemic damage to the liver parenchyma.

Ultrasonic shears were used for the resection of superficial liver parenchyma. A Cavitron Ultrasonic Surgical Aspirator (CUSA; Integra Lifesciences, Plainsboro, NJ, USA) was used for deeper dissection. Bleeding from minor branches of the hepatic vein was controlled by an endoclip and sealing device, while that from the large or main hepatic branches was controlled directly by suture. Specimen extraction was performed through the extension of the subumbilical port site. We reduced the chance of further bleeding by performing meticulous irrigation and hemostasis with fibrin glue.

### 2.4. Statistics

All statistical analyses were performed using SPSS software version 22.0 (IBM Corp., Armonk, NY, USA). Data were reported as the median (range). Categorical variables were compared using Fisher’s exact *t*-test and continuous variables were compared using the nonparametric Mann–Whitney U test. Survival rates were analyzed using the Kaplan–Meier method and compared using log-rank tests. *p*-values < 0.05 were considered statistically significant. The study was approved by the institutional review board of Seoul National University Bundang Hospital (Study approval number: B-2109-708-105).

## 3. Results

The preoperative characteristics of the LLR and OLR groups are summarized in Table 1. The median tumor size was significantly smaller in the LLR group than in the OLR group (3.0 vs. 5.0 cm; *p* < 0.001). The portion of female patients was greater in the LLR group than in the OLR group (30.5% vs. 13.7%; *p* = 0.036). There were no significant differences between the two groups in terms of age, body mass index (BMI), presence of hepatitis, rate of preoperative radiofrequency ablation, and transarterial chemoembolization.

The perioperative outcomes are summarized in Table 2. Operation time was similar in both groups. However, the blood loss (500 vs. 700 mL; *p* < 0.001) and transfusion rate (10.2 vs. 31.4%; *p* = 0.006) were significantly lower in the LLR group than in the OLR group. In the LLR group, most of the patients underwent segmentectomy (47.5%), followed by right hemihepatectomy (10.2%) and right anterior sectionectomy (8.5%). In the OLR group, right hemihepatectomy (23.5%) was most frequently performed, followed by central bisectionectomy (15.7%) and right anterior sectionectomy (11.8%).

The pathologic and postoperative outcomes are shown in Table 3. Resection margin, portion of patients with liver cirrhosis, number of satellite nodules, and microvascular invasion were similar in both groups. The rate of R0 resection was higher in the LLR group than in the OLR group (100% vs. 92.2%; *p* = 0.043). There were no significant differences in overall complication rates between both groups. However, hospital stay was significantly shorter in the LLR group than in the OLR group (6 vs. 10 days; *p* < 0.001). There was no mortality within 3 months in neither group.

There was no significant difference in the 5-year overall survival rate (100% vs. 75.7%; *p* = 0.052) and 5-year disease-free survival rate (65.6% vs. 41.6%; *p* = 0.076) between both groups (Figure 1).

To account for the improvements of advanced techniques and devices, patients in the LLR group were further divided into two groups according to the date of operation, Group 1 (n = 19) and Group 2 (n = 40), who underwent LLR before and after 2015, respectively (Table 4). Hospital stay was significantly shorter in Group 2 than in Group 1 (6 vs. 8 days; *p* = 0.006). Shorter median operation time (280 vs. 360 min; *p* = 0.036), less blood loss (455 vs. 500 mL; *p* = 0.075), and lower rate of transfusion (5.0% vs. 21.1%; *p* = 0.078) were observed in Group 2 compared with Group 1, although the differences were not significant.

## 4. Discussion

In 2013, Yoon et al. reported the first case series that demonstrated the feasible application of laparoscopic resection for centrally located tumors [12]. With the accumulation of experience and improvement of the laparoscopic techniques, we attempted to extend the study by comparing the long-term outcomes with the open resection group.

In this study, we compared the perioperative outcomes of LLR and OLR for cHCC. We have shown that blood loss, transfusion rate, and hospital stay were significantly better in the LLR group. Regarding survival analysis, the 5-year overall survival and 5-year disease-free survival was similar in both groups. To account for the adaptation of technical advancement and introduction of new devices, we further divided the LLR group into two subgroups, Group 1 and Group 2, who underwent LLR before and after 2015, respectively. Extensive utilization of the new techniques showed a shorter hospital stay. Operation time, blood loss, and transfusion rate tended to improve in the recent LLR group, although we failed to show statistical significance.

LLR for minor resection and left lateral sectionectomy is widely accepted as a standard treatment option for HCC [10]. However, LLR for cHCC is extremely challenging for several reasons, including difficulty in exposing surgical field, controlling bleeding, and acquiring adequate resection margins [12,13,14]. Most types of resections for cHCC require major hepatectomy to achieve adequate resection margins. For major LLR, the current consensus is that it should be performed only at a high-volume medical center with abundant experience in laparoscopic surgery [10,11].

To overcome such difficulties, our institution has utilized novel techniques. We introduced an additional trocar at the intercostal spaces for better visualization of tumors. Thus, the laparoscopic instruments could reach longer distances and help surgeons to easily access the operative field. The advantages of such a technique have been demonstrated in previous study [17]. We have also adopted the semi-lateral French position with a 30° reverse Trendelenburg position for better visualization of the liver hilum and exposure of the right posterior segments.

Another obstacle to LLR for cHCC is the difficulty in controlling bleeding. Thus, we introduced the laparoscopic Pringle’s maneuver to provide better bleeding control. Previous studies have demonstrated the advantages of the Pringle maneuver in laparoscopic operation [22,23]. The utilization of laparoscopic CUSA also helped surgeons avoid injuries to major hepatic veins and made more meticulous dissection around the hilum possible. If bleeding occurred, we directly applied a laparoscopic endoclip for small vessels. For major vessel injury, we applied direct suture. These techniques led to less blood loss and lower rate of transfusion.

Additionally, achieving adequate resection margins is challenging in LLR for cHCC since these tumors are located very close to the hilum or major vascular structures. Acquiring a sufficient resection margin is important since it may affect the overall survival of the patients and recurrence rate [24,25]. The utilization of laparoscopic ultrasonography helped identify the exact tumor location and obtain the adequate margin. We demonstrated that the resection margin for both OLR and LLR groups was similar. However, the rate of R0 resection was slightly lower in the OLR group compared with the LLR group (100% vs. 92.2%; *p* = 0.043). Furthermore, the 5-year disease-free survival rate was lower in the OLR group than in the LLR group. This may be due to selection bias for the OLR group since we preferred open resection for tumors greater than 5 cm. In addition, the number of complex resections, such as central bisectionectomy and right anterior sectionectomy, was higher in the OLR group than in the LLR group, which may have further hindered obtaining adequate resection margins. However, since HCC is a systemic disease, merely extending the resection margin may not be sufficient to prevent recurrence. Therefore, in complex surgical resection, an individualized approach must be taken considering the patient’s condition and function of the remnant liver [15].

For the survival analysis, 5-year overall survival rate appeared to be improved in the LLR group although it was not statistically significant. This may be explained by the higher R0 resection rate and smaller tumor size in the LLR group.

There were some limitations to this study. First, this is a nonrandomized, retrospective, single-center study with a relatively small sample size. Second, there were some differences in the preoperative characteristics, such as tumor size, CTP, PT INR, and albumin, between the two groups. These factors can affect postoperative morbidity and mortality. Despite these differences, propensity score matching was inevitably impossible due to the small sample size. Third, since the study period is over 10 years, this may be associated with a bias. Thus, further randomized, larger cohort studies are warranted to validate the advantages of LLR over OLR for cHCC.

## 5. Conclusions

In conclusion, this study suggests that LLR can be safely performed for cHCC in highly selected patients. However, it must be performed at specialized, high-volume centers with surgeons with sufficient experience in laparoscopic resection. We believe that future improvements and development in the laparoscopic techniques and devices will further extend the indication of LLR to tumors at more complex locations.

## Figures and Tables

**Figure 1 medicina-58-00737-f001:**
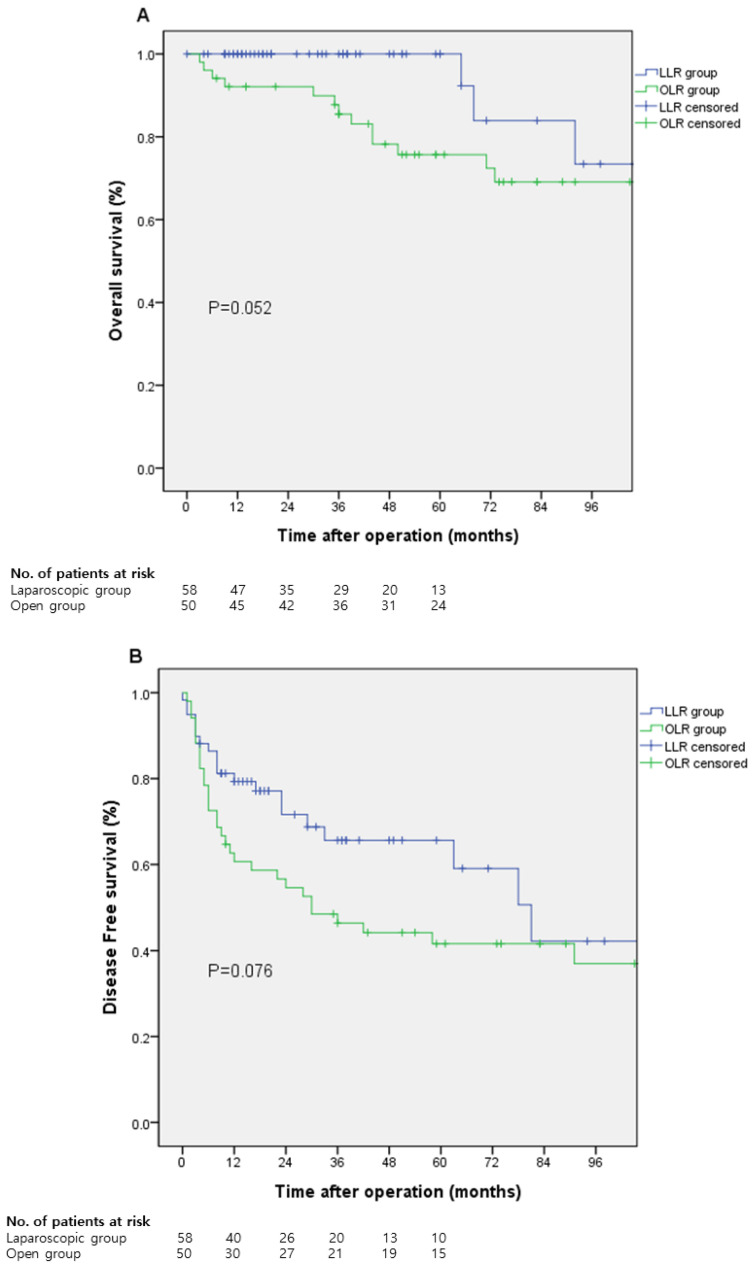
Survival curve. (**A**) the 5-year overall survival rate, (**B**) 5-year disease-free survival rate.

**Table 1 medicina-58-00737-t001:** Preoperative characteristics of laparoscopic and the open groups for centrally located tumors.

	LLR (*n* = 59)	OLR (*n* = 51)	*p*-Value
Age, (years), median (range)	57 (26–74)	57 (30–85)	0.926
Gender			0.036
Male	41 (69.5)	44 (86.3)	
Female	18 (30.5)	7 (13.7)	
BMI (kg/m^2^), median (range)	24.4 (16.36–31.61)	24.2 (16.73–32.06)	0.242
Tumor size (cm), median (range)	3.0 (0.9–10.3)	5.0 (1.5–13.0)	0.000
Location of tumor			
Segment 1	7 (11.9)	2 (3.9)	
Segment 4	15 (25.4)	11 (21.6)	
Segment 5	10 (16.9)	14 (27.5)	
Segment 8	15 (25.4)	11 (21.6)	
Segment 1 and 8	0	1 (2.0)	
Segment 4 and 5	1 (1.7)	2 (4.0)	
Segment 4 and 8	2 (3.4)	6 (11.8)	
Segment 5 and 8	6 (10.2)	3 (5.9)	
Segment 4, 5, 8	3 (5.1)	1 (2.0)	
Albumin (g/dL), median (range)	4.3 (1.3–4.9)	4.1 (2.5–5.1)	0.011
Bilirubin (mg/dL), median (range)	0.7 (0.2–2.4)	0.8 (0.3–2.8)	0.149
PT-INR, median (range)	1.05 (0.9–1.24)	1.1 (0.9–1.45)	0.005
PLT (1000/µL), median (range)	179 (73–334)	176 (38–424)	0.590
SGOT (IU/L), median (range)	37.0 (14–176)	36 (20–118)	0.563
SGPT (IU/L), median (range)	33.0 (11–256)	36 (7–260)	0.500
AFP (ng/mL), median (range)	7.6 (1.2–6540)	16.0 (1.4–35,000)	0.096
Child Pugh class, n (%)			0.002
A	58 (98.3)	40 (78.4)	
B	1 (1.8)	7 (13.7)	
C	0	4 (7.8)	
Hepatitis, n (%)			1.000
Hepatitis B	44 (74.6)	38 (74.5)	
Hepatitis C	3 (5.4)	2 (3.9)	
Both positive	0	0	
Both negative	12 (20.3)	11 (21.6)	
Prior RFA, n (%)	4 (6.9)	1 (2.0)	0.369
Prior TACE, n (%)	13 (22.4)	17 (33.3)	0.203

LLR: laparoscopic liver resection, OLR: open liver resection, BMI: body mass index, PT: prothrombin time, INR: international normalized ratio, PLT: platelet count, SGPT: serum glutamic pyruvic transaminase, SGOT: serum glutamic-oxaloacetic transaminase, AFP: alpha-fetoprotein, RFA: radiofrequency ablation, and TACE: transarterial chemoembolization.

**Table 2 medicina-58-00737-t002:** Perioperative outcomes.

	LLR (*n* = 59)	OLR (*n* = 51)	*p*-Value
Operation type, n (%)			0.013
Caudate lobectomy	5 (8.5)	3 (5.9)	
Segmentectomy	28 (47.5)	9 (17.6)	
Bi-segmentectomy	2 (3.4)	2 (3.9)	
Extended segmentectomy	3 (5.1)	0	
Left hemihepatectomy	4 (6.8)	5 (9.8)	
Right anterior sectionectomy	5 (8.5)	6 (11.8)	
Right posterior sectionectomy	2 (3.4)	1 (2.0)	
Right hepatectomy	6 (10.2)	12 (23.5)	
Extended right hepatectomy	1 (1.7)	4 (7.8)	
Central bisectionectomy	3 (5.1)	8 (15.7)	
Operation time (min), median (range)	285 (70–790)	280 (105–745)	0.938
Blood loss (mL), median (range)	500 (10–5900)	700 (150–7000)	0.000
Transfusion, n (%)	6 (10.2)	16 (31.4)	0.006

LLR: laparoscopic liver resection; OLR: open liver resection.

**Table 3 medicina-58-00737-t003:** Pathologic and postoperative outcomes.

	LLR (*n* = 59)	OLR (*n* = 51)	*p*-Value
Resection margin (cm), median (range)	0.4 (0.0–5.0)	0.5 (0.0–3.5)	0.398
Cirrhosis, n (%)	35 (59.3)	30 (58.8)	1.000
Satellite nodule, n (%)	3 (5.1)	3 (5.9)	1.000
Microvascular invasion, n (%)	30 (51.7)	25 (49.0)	0.778
Resection, n (%)			0.043
R0	59 (100)	47 (92.2)	
R1	0	4 (7.8)	
Postoperative complication, n (%)	13 (22.0)	14 (27.5)	0.510
Complication type, n (%)			0.319
General	3 (5.1)	2 (3.9)	
Surgical	7 (11.9)	5 (9.8)	
Liver related	3 (5.1)	3 (5.9)	
Mixed	0	4 (7.8)	
Clavien–Dindo grade, n (%)			0.798
I	2 (3.4)	1 (2.0)	
II	1 (1.7)	3 (5.9)	
IIIa	6 (10.2)	7 (13.7)	
IIIb	4 (6.8)	3 (5.9)	
IVa	0	0	
IVb	0	0	
V	0	0	
Hospital stay (days), median (range)	6 (2–59)	10 (4–64)	0.000
Mortality within 3 months, n (%)	0	0	

LLR: laparoscopic liver resection; OLR: open liver resection.

**Table 4 medicina-58-00737-t004:** Comparison of perioperative outcomes between patients who underwent LLR before and after 2015.

	Group 1 (*n* = 19)	Group 2 (*n* = 40)	*p*-Value
Operation type, n (%)			0.264
Segmentectomy	8 (42.1)	23 (57.5)	
Bi-segmentectomy	0	2 (5.0)	
Left hemihepatectomy	2 (10.5)	2 (5.0)	
Right anterior sectionectomy	1 (5.3)	4 (10.5)	
Right posterior sectionectomy	2 (10.5)	0	
Right hepatectomy	4 (21.1)	2 (5.0)	
Extended right hepatectomy	0	1 (2.5)	
Caudate Lobectomy	1 (5.3)	4 (10.0)	
Central bisectionectomy	1 (5.3)	2 (5.0)	
Operation time (min), median (range)	360 (80–790)	280 (70–755)	0.036
Blood loss (mL), median (range)	500 (200–5900)	455 (5–2000)	0.075
Transfusion, n (%)	4 (21.1)	2 (5.0)	0.078
Hospital stay (days), median (range)	8 (4–59)	6 (2–19)	0.006

Group 1 patients who underwent LLR before the introduction of new techniques in 2015 and Group 2 patients who underwent LLR after the introduction of new techniques in 2015; LLR: laparoscopic liver resection.

## Data Availability

The confidentiality rules of hospital.

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
