# Peer review of "Long-Term Outcomes of Laparoscopic Liver Resection for Centrally Located Hepatocellular Carcinoma"

_medicina, 2022, doi:10.3390/medicina58060737_

Round 1
Reviewer 1 Report
The article Long-term outcomes of laparoscopic liver resection for centrally located hepatocellular carcinoma endorse the feasibility and also the advantages of the laparoscopy in liver resections.
Altough the data are nicely presented and the results very well underlined there is a minor issue that has to be cleared out: in the group of 59 patients that underwent laparoscopic liver resections there were two conversion to open surgery and these patients data were excluded but in the tables there are still 59 patients and the statistic calculation seemed to be done on the entire group for 59 instead of 57.
Another minor comment in order to ease the data interpretation: at lines 167-170 when results on the two laparoscopic subgroups were presented, I would suggest to change the order of the numbers between the brackets so they correspond to the order in which group 2 and 1 are in the text.
Author Response
Reviewer 1.
The article Long-term outcomes of laparoscopic liver resection for centrally located hepatocellular carcinoma endorse the feasibility and also the advantages of the laparoscopy in liver resections.
Although the data are nicely presented and the results very well underlined there is a minor issue that has to be cleared out: in the group of 59 patients that underwent laparoscopic liver resections there were two conversion to open surgery and these patients data were excluded but in the tables there are still 59 patients and the statistic calculation seemed to be done on the entire group for 59 instead of 57.
- Thank you for your important comments. Laparoscopic liver resection for tumors located in central portion is a technically demanding procedure. Therefore, the open conversion is a major issue when discussing the safety and feasibility of the laparoscopic liver resection. Therefore, we included the patients who underwent open conversion to evaluate the clinical outcomes of procedures.
Another minor comment in order to ease the data interpretation: at lines 167-170 when results on the two laparoscopic subgroups were presented, I would suggest to change the order of the numbers between the brackets so they correspond to the order in which group 2 and 1 are in the text.
- Thank you for your comments. As your recommendation, we revised the manuscript.

Reviewer 2 Report
Dear Author,
We read with great attention the manuscript entitled “Long-term outcomes of laparoscopic liver resection for centrally located hepatocellular carcinoma”.
This manuscript aims to investigate the short-term and long-term results of laparoscopic resection of central Hepatocellular carcinoma (cHCC).
The authors already published on feasibility of laparoscopic approach for cHCC and this series’ purpose is to investigate the short-term and long-term safety of this approach.
The authors performed more than 50 laparoscopic liver resections for HCC a year which mean they are in a very high liver surgery volume but the major issue is that there was no matching on the most important predictive factors of postoperative morbidity and mortality as well as the prognostic factor between the laparoscopic and open cases.
As a result, both group laparoscopic and open are very different for example on the rates of Child B and C cirrhosis and are not fit for comparison. Indeed, the external validation of this study in term of comparative postoperative outcomes and survival are very questionable.
Thus, the authors should explain that when tumor size greater than 5 cm in diameter and tumors invading or adjacent to the main portal pedicle or IVC the laparoscopic approach was not proposed and that only “easy” cHCC were included in the laparoscopic retrospective series.
Comparable survival rates analysis is not possible in thus situation.
The design of the study is not sufficiently robust to authorize the authors conclusions and this series offers new data on a rarely investigated subject.
The manuscript redaction should be improved as well as English redaction.
Author Response
We read with great attention the manuscript entitled “Long-term outcomes of laparoscopic liver resection for centrally located hepatocellular carcinoma”. This manuscript aims to investigate the short-term and long-term results of laparoscopic resection of central Hepatocellular carcinoma (cHCC). The authors already published on feasibility of laparoscopic approach for cHCC and this series’ purpose is to investigate the short-term and long-term safety of this approach.
The authors performed more than 50 laparoscopic liver resections for HCC a year which mean they are in a very high liver surgery volume but the major issue is that there was no matching on the most important predictive factors of postoperative morbidity and mortality as well as the prognostic factor between the laparoscopic and open cases.
As a result, both group laparoscopic and open are very different for example on the rates of Child B and C cirrhosis and are not fit for comparison. Indeed, the external validation of this study in term of comparative postoperative outcomes and survival are very questionable. Thus, the authors should explain that when tumor size greater than 5 cm in diameter and tumors invading or adjacent to the main portal pedicle or IVC the laparoscopic approach was not proposed and that only “easy” cHCC were included in the laparoscopic retrospective series. Comparable survival rates analysis is not possible in thus situation. The design of the study is not sufficiently robust to authorize the authors conclusions and this series offers new data on a rarely investigated subject.
The manuscript redaction should be improved as well as English redaction.
- Thank you for your important comments. As your comments, it is a main limitation of our study. There were some differences in the preoperative characteristics, such as tumor size, child-pugh score, PT INR and albumin, between two groups. These factors can affect the postoperative morbidity and mortality. Despite these differences, propensity score matching was inevitably impossible due to the small sample size. We revised the “limitation” and “conclusion” section.
Before revision (Discussion_Limitation)
There were some limitations to this study. First, this is a nonrandomized, retrospec-tive, single-center study with a relatively small sample size. Second, there were some dif-ferences in the preoperative characteristics between the two groups. For example, the tu-mor size was greater in the OLR group than in the LLR group. Third, since the study pe-riod is over 10 years, this may be associated with a bias. Thus, further randomized, larger cohort studies are warranted to validate the advantages of LLR over OLR for cHCC.
After revision (Discussion_Limitation)
There were some limitations to this study. First, this is a nonrandomized, retrospec-tive, single-center study with a relatively small sample size. Second, there were some differences in the preoperative characteristics, such as tumor size, CTP, PT INR and albumin, between two groups. These factors can affect the postoperative morbidity and mortality. Despite these differences, propensity score matching was inevitably impossible due to the small sample size. Third, since the study pe-riod is over 10 years, this may be associated with a bias. Thus, further randomized, larger cohort studies are warranted to validate the advantages of LLR over OLR for cHCC.
Before revision (Conclusion)
In conclusion, this study suggests that LLR can be safely performed for cHCC in se-lected patients.
After revision (Conclusion)
In conclusion, this study suggests that LLR can be safely performed for cHCC in highly selected patients.
